# Application of Environment-Friendly Rhamnolipids against Transmission of Enveloped Viruses Like SARS-CoV2

**DOI:** 10.3390/v13020322

**Published:** 2021-02-20

**Authors:** Ling Jin, Wendy Black, Teresa Sawyer

**Affiliations:** 1Department of Biomedical Sciences, Carlson College of Veterinary Medicine, Oregon State University, Corvallis, OR 97331, USA; wendy.black@oregonstate.edu; 2Department of Microbiology, College of Science, Oregon State University, Corvallis, OR 97331, USA; 3Electron Microscopy Facility, 145 Linus Pauling Science Center, Oregon State University, Corvallis, OR 97331, USA; SawyerT@Science.oregonstate.edu

**Keywords:** rhamnolipids, 222A, 222B, HSV-1, HSV-1-GFP, coronavirus, bovine coronavirus

## Abstract

In the face of new emerging respiratory viruses, such as SARS-CoV2, vaccines and drug therapies are not immediately available to curb the spread of infection. Non-pharmaceutical interventions, such as mask-wearing and social distance, can slow the transmission. However, both mask and social distance have not prevented the spread of respiratory viruses SARS-CoV2 within the US. There is an urgent need to develop an intervention that could reduce the spread of respiratory viruses. The key to preventing transmission is to eliminate the emission of SARS-CoV2 from an infected person and stop the virus from propagating in the human population. Rhamnolipids are environmentally friendly surfactants that are less toxic than the synthetic surfactants. In this study, rhamnolipid products, 222B, were investigated as disinfectants against enveloped viruses, such as bovine coronavirus and herpes simplex virus 1 (HSV-1). The 222B at 0.009% and 0.0045% completely inactivated 6 and 4 log PFU/mL of HSV-1 in 5–10 min, respectively. 222B at or below 0.005% is also biologically safe. Moreover, 50 μL of 222B at 0.005% on ~1 cm^2^ mask fabrics or plastic surface can inactivate ~10^3^ PFU HSV-1 in 3–5 min. These results suggest that 222B coated on masks or plastic surface can reduce the emission of SARS-CoV2 from an infected person and stop the spread of SARS-CoV2.

## 1. Introduction

Enveloped viruses, such as coronaviruses, influenza viruses, and herpesviruses, are all wrapped with a phospholipid bilayer called the envelopes. Viral envelopes are cellular phospholipids, which are taken from the infected cell during the budding process. Surfactants are amphiphilic organic compounds comprised of a hydrophobic non-polar hydrocarbon tail and a hydrophilic polar head. They possess an aqueous solubility in the micro- or millimolar range [1]. The compatibility of the hydrophobic tails with phospholipid bilayer allows surfactants to penetrate through the membrane easily in small aqueous range, consequently change the conformation of phospholipid bilayer membrane or the envelope, and separate the membrane lipids. Therefore, the interaction between the phospholipid bilayer and the surfactant has detrimental effect on integrity of enveloped virions (complete viral particle). Enveloped viruses have glycoproteins on their surfaces which act as host receptor binding proteins. If enveloped viruses lose their envelopes, they will lose their glycoproteins. Therefore, viruses without envelopes will not bind to the host receptors and infect their target cells. If the integrity of the viron’s envelopes is lost, there is a loss in ability to enter the target cells, and subsequent loss of infectivity. Therefore, the surfactants can be used to kill or inactivate enveloped viruses.

Commercial bleach products, such as Clorox, Lysol, are commonly used to disinfect enveloped viruses in research labs and medical facilities. They are synthetic detergents or surfactants and can cause skin irritation or harm the environment. Rhamnolipids are biosurfactants, which are of biological origin and have many advantages over synthetic counterparts, such as low toxicity and high biodegradability [2,3,4]. Rhamnolipids were first identified in 1949 and purified in 1968 [5,6]. Since then, numerous microorganisms, including bacteria, fungi, and yeast, have been reported to produce rhamnolipids [7,8,9]. The rhamnolipid structure is determined by the number of rhamnoses (one or two), fatty acids (one or two) and the fatty acid composition. Therefore, rhamnolipids are commonly classified into two groups: Monorhamnolipids and dirhamnolipids. Rhamnolipids have been used as an emulsifier, stabilizer, solubilizer, wetting, foaming agent, bactericide and fungicide [10,11,12,13,14,15,16,17,18].

Similar to SARS-CoV2, bovine coronavirus (BCoV) is a member of the *Coronaviridae* family, which comprises enveloped positive-sense single-stranded RNA viruses. BCoV is a pneumoenteric virus that infects the upper and lower respiratory tract, and intestine. Infection of BCoV can cause calf diarrhea, winter dysentery, and respiratory infections in cattle of various ages [19]. Human herpesvirus type 1 (HSV-1) is a member of the *Herpesviridae* family that include a large number of enveloped DNA viruses common in both animal and humans. HSV-1 is the cause of cold sores or fever blisters in or around the mouth and encephalitis in newborns [20,21]. It is also the leading cause of corneal blindness in developing countries [22,23]. Both are enveloped viruses, and their surface proteins on the envelopes are essential for virus entry in infection. Rhamnolipids have been shown to be capable of inactivating both bacteria and fungi [24,25,26]. In this study, the ability of rhamnolipids to inactivate BCoV and HSV-1 was investigated.

## 2. Materials and Methods

### 2.1. Surfactants

Rhamnolipids are provided by AGAE Technologies, LLC (Corvallis, Oregon, USA). AGAE0222A is a 10% aqueous solution made with commercially available di-rhamnolipid predominant product R95Dd. AGAE0222B is a 10% aqueous solution made with commercially available mono-rhamnolipid predominant product R95Md. Both rhamnolipid solutions were adjusted to pH7.4 ± 0.1. In this study, rhamnolipids AGAE0222A and AGAE0222B are called 222A, and 222B, respectively. Sodium dodecyl sulfate (SDS) (EM-7910) and Triton X-100 (X-100) were purchased from OmniPur and Sigma-Aldrich, respectively.

### 2.2. Viruses and Cells

HSV-1 McKrae [27] and HSV-1-GFP [28] were cultured in Vero cells in Eagle’s Minimum Essential Medium (EMEM) supplemented with 5% fetal bovine serum (FBS) (GeminiBio, Sacramento, CA, USA), penicillin (100 U/mL), and streptomycin (100 μg/mL) (Sigma-Aldrich, Inc., St. Louis, MO, USA). Vero cells were cultured in EMEM supplemented with 10% fetal bovine serum (FBS) (GeminiBio, USA), penicillin (100 U/mL), and streptomycin (100 μg/mL) (Sigma-Aldrich, Inc., St. Louis, MO, USA) at 37 °C with 5% CO_2_ in a humidified incubator as we described previously [22,29].

Bovine coronavirus (BCoV) was the NADL-Nebraska Strain isolated by the National Veterinary Services Laboratories (NVSL) in 1981 [30]. BCoV was grown on human rectal tumor (Hrt-18G) cells [30,31], which were maintained in EMEM supplemented with 10% FBS (GeminiBio, USA), penicillin (100 U/mL) and streptomycin (100 μg/mL) (Sigma-Aldrich, Inc.) at 37 °C with 5% CO_2_ in a humidified incubator. The virus was propagated in EMEM supplemented with 2.5 μg/mL trypsin and 2.5 μg/mL pancreatin, 1× insulin–transferrin–selenium (Cat No. 51500-056, GIBCO) [31].

### 2.3. Rhamnolipid Cytotoxicity Assay In Vitro

Ninety-six-well plates were seeded with ~2 × 10^4^ Vero cells or Hrt-18G cells per well and grown overnight at 37 °C and 5% CO_2_. The cells were washed once with phosphate-buffered saline (PBS) and treated in 100 μL 222A or 222B with indicated concentrations diluted in PBS at 37 °C for 1h. The rhamnolipid-treated cells were then washed once with PBS and then replenished with EMEM containing 5% FBS (GeminiBio, USA) and antibiotics (as described above), and further incubated for 24 h. After incubation, cell morphology was examined under an inverted light microscope, and cell viability was examined by using colorimetric cell viability kit III (XTT) (PromoKine, Huissen, Netherlands). Briefly, 50 μL of the XXT reaction solution was added to each well, and the plate was incubated at 37 °C incubator for 3–4 h. Absorbance at a wavelength of 450–500 nm was read on a microplate fluoresce reader (BioTek, Winoosk, VT, USA) and recorded. Wells containing 222A or 222B at each concentration in media without cells served as a blank to ensure that rhamnolipids themselves were not registering fluorescence [22]. In addition, the toxicity of SDS and Triton-X100 was examined similarly as 222B.

### 2.4. HSV-1 and BCoV Infection In Vitro in the Presence or Absence of Rhamnolipids

Twelve-well plates were prepared the day before infection by seeding each well with approximately 1 × 10^5^ Vero cells or Hrt18G cells. The cells were washed once with PBS and infected with 100 μL of either ~2 × 10^7^ PFU/mL of HSV-1-GFP or ~1 × 10^7^ TCID_50_/_mL_ of BCoV in the presence or absence of rhamnolipids. Following 1-hr absorption, the cells were replenished with 1 mL culture medium containing 5% FBS and antibiotics (as described above) in the presence or absence of 222B at the indicated concentration. At 24 or 72 h post-infection (hpi), the cytopathic effect (CPE) from virus infection was examined under an inverted light microscope or fluorescent microscope.

### 2.5. Immunoflorescence Staining

Hrt-18G cells infected with BCoV were washed with ice-cold PBS twice and fixed in 1 mL ice-cold methanol at −20 °C for 20 min. Following fixation, the plates were washed again twice with cold PBS and blocked in 1% bovine serum albumin (BSA) in Tris-buffered saline (TBS) at room temperature for 1 h, then stained with 1:40 diluted anti-BCoV monoclonal antibody conjugated with Fluorescein isothiocyanate (FITC) at room temperature for 1 h on a rocker and washed twice with PBS before viewing under fluoresce microscope.

### 2.6. Electron Microscopy of Virions

Six 75-cm^2^ flasks seeded with 90% confluent Vero cells were infected with 0.01M.O.I. HSV-1-McKrae. The infected flasks were harvested when 80–90% CPE developed at 3–4 days post-infection (dpi), and subjected to two-freeze and thaw cycles between −80 °C and room temperature. The supernatant was cleared of cells and cell debris by centrifugation at 9000× *g* for 30 min at 4 °C. The virions were then centrifuged through a 60% sucrose cushion in a Beckman Model XL-70 ultracentrifuge at 25,000 rpm for 1h in a SW28 rotor at 4 °C, as we described previously [32]. The virion pellets were suspended in 100 μL dH2O first, and then 10 μL of virion was mixed with 90 μL of 0.01% SDS, 0.01% Triton X-100, and 0.01% 222B, respectively. The treated virions were stained as we described previously [32]. Briefly, treated virion suspensions were adsorbed to formvar-coated carbon-stabilized copper grids by floating grids on ~20 μL drops of the sample spotted on parafilm. The grids were then blotted dry with Whatman filter paper and immediately floated on ~20 μL drops of 2% phosphotungstic acid (PTA) (pH 6.9) in water for 30 s. Excess PTA was removed by side-blotting and the grids were allowed to air dry [32]. Images were obtained with a FEI Titan 80–200/ChemiSTEM Transmission Electron Microscope (TEM).

### 2.7. HSV-1 Plaque Reduction Assay Following Rhamnolipid Treatment

Vero cells were seeded in 12-well tissue culture plates at approximately 2 × 10^5^ cells/well on the day before infection. Three wells were infected with HSV-1 at different concentrations, ranging from 1 × 10^3^ to 1 × 10^7^ PFU/mL, in the presence or absence of rhamnolipids, 222B. Following a 1-h viral absorption, 3 mL of 2% methylcellulose overlay media was added to each well. Plates were incubated at 37 °C with 5% CO_2_ in a humidified incubator for 4–5 days. The plates were fixed in 20% methanol, stained with 1% crystal violet, plaques were counted as described previously [22].

### 2.8. Plastic Surface Coating with 222B and Surface-Contact Assay

The 48-well Corning Costar Flat Bottom Cell Culture Plate with a growth area of 0.95 cm^2^ was selected for 222B coating on the plastic surface. The bottom of the well was coated by adding 20 μL of 222B at 0.005% directly, which covers the entire surface of the well. The coated wells were air-dried overnight. Virus droplets in 10 μL containing 10^3^ or 10^4^ virions (PFU) of HSV-1 were added to the bottom of the well for 1 min or 5 min. The contact between HSV-1 and 222B coated plastic surface was stopped by adding 1 mL of PBS elution buffer. Viruses left on the plastic surface were eluted in 1 mL PBS for 1 h at room temperature on a plate rocker. The control wells were coated with PBS only. The non-surface contact controls were 222B coated wells, the viruses in 10 μL were added to the coated wells after application of 1 mL PBS elution buffer, therefore, had no direct contact with the coated surface. 250 μL of eluted viruses from each well were then titrated by standard plaque assay.

### 2.9. Fabric Coating with 222B and Surface-Contact Assay

Approximately 1 cm^2^ square fabrics were cut from a commercial surgical mask, made of non-woven fabric (Tronex Healthcare, Mount Olive, NJ, USA), and then soaked in 50 μL of 0.005% 222B or PBS. The soaked fabrics were air-dried overnight after removing the excess liquid from the soaked fabrics. The dried fabrics were then placed in a well of 24-well plate (Corning, Glendale, AZ, USA). Virus droplets in 10 μL containing 10^3^ or 10^4^ PFU HSV-1 were added to the coated fabrics for 3–5 min. The contact between viruses and 222B coated fabrics was stopped by adding 1 mL PBS elution buffer directly to the well. Viruses left on the fabric surface were eluted at room temperature for 1h on a plate rocker. The non-surface contact controls were 222B coated fabric, the viruses in 10 μL were added onto the fabrics after application of 1 mL PBS elution buffer, therefore, had no direct contact with the coated fabrics. The mock controls were viral droplets added directly to 1 mL PBS in wells that were not coated. The eluted viruses were then titrated by standard plaque assay.

### 2.10. Statistics

All statistical analyses were performed using GraphPad Prism version 4 for Windows (GraphPad Software, San Diego, CA, USA). Cell viability results and plaque reduction results were analyzed by two-way ANOVA with Bonferroni post tests.

## 3. Results

### 3.1. The Rhamnolipid Treatment Effect on HSV-1 and Coronavirus (BCoV) Replication In Vitro

The antiviral effect of rhamnolipids, monorhamnolipids (222B) and dirhamnolipids (222A) were first examined against HSV-1. About 2 × 10^7^ PFU/mL of HSV-1 was diluted (10 fold) in PBS containing 0.009% or 0.001% of 222B or 222A first, then inoculated to a 12-well plate seeded with Vero cells. Vero cells are susceptible to HSV-1 infection in vitro and produce visible morphology change (cytopathic effect) at 24 h post-infection. If the rhamnolipids can kill the virus, then no cytopathic effect (CPE) will be produced in HSV-1 infected cells. Figure 1 shows the cells infected with HSV-1 in the presence or absence of rhamnolipids. In the absence of rhamnolipids, 90% cytopathic effect (CPE) was observed at 24 h post-infection (hpi) (Figure 1, HSV-1 only). In the presence of 0.001% 222A or 222B, 80–90% CPE was also observed, which was similar to infection in HSV-1 only at 24 hpi. In the presence of 0.009% 222A, only 20–30% CPE was visible at 24 hpi (HSV-1+222A). However, in the presence of 0.009% 222B (HSV-1+222B), no CPE was observed at 24 hpi, which is similar to cells without infection at 24 hpi. This suggests that rhamnolipids have antiviral effects against enveloped viruses HSV-1. Since 0.009% 222B had better protection against HSV-1 infection than 222A did, 222B is selected for the remaining study.

To prove the rhamnolipid’s anti-viral activity is not HSV-1 specific but specific to enveloped viruses, 222B was tested against bovine coronaviruses (BCoV) infection in vitro. Hrt-18G cells are susceptible to BCoV infection in vitro and produce visible morphology change (cytopathic effect) at 72 h post-infection. If the rhamnolipids can kill the virus, then no cytopathic effect (CPE) will be produced in BCoV infected cells. Hrt-18G cells in a 12-well plate were infected with BCoV at ~1 × 10^6^ TCID_50_/well in the presence of 0.009% or 0.001% of 222B. At 72 hpi, BCoV CPEs, such as cell rounding, inclusion bodies (black arrows), were visible in cells infected with BCoV without 222B (Figure 2A, BCoV only). However, in the presence of 222B at 0.009%, no CPE was visible in cells infected with BCoV (BCoV+0.009%). To further confirm the replication of BCoV, infected cells were stained with FITC labeled mAb specific to BCoV at 72 h post-infection. As shown in Figure 2B, positive FITC staining (white arrows) was only visible in wells with BCoV only and BCoV infection with 0.001% 222B. In contrast, no FITC staining was visible in wells infected with BCoV containing 0.009% 222B. Noticeably, in the presence of 0.001% 222B, positive BCoV staining (Green fluoresce, white arrows) is significantly reduced compared to the cells with BCoV only. This result suggests that rhamnolipids have antiviral effects against enveloped viruses, such as HSV-1 and BCoV.

### 3.2. Cytotoxicity of Rhamnolipids In Vitro

To fully understand how rhamnolipids inactivate the enveloped viruses, we need to ensure that the concentration of rhamnolipids used in our study has no cytotoxicity to the cells. Only viable cells can support the virus replication in vitro. Therefore, cell cultures are exposed to different concentrations of rhamnolipid for 60 min in the cytotoxicity assay. To determine whether rhamnolipids affect cell viability, Vero cell and Hrt-18G cell viability were tested under different concentrations of rhamnolipids by colorimetric cell viability kit III (XTT). The assay is based on the ability of mitochondria enzyme from metabolic active cells to reduce the tetrazolium salt XTT to orange-colored formazan compounds. The dye formed is water-soluble, and its intensity is proportional to the number of metabolically active cells. The greater the number of active cells in the well, the greater the activity of mitochondria enzymes, the higher the concentration of the dye formed, the higher absorbance on the ELISA reader. As shown in Figure 3A, Vero cells treated with rhamnolipids, 222A and 222B, at or above 0.05%, had less than 10% viable cells left, compared to the untreated cells (Mock-treated). Cells treated with rhamnolipids 222A at 0.01% and 0.009% were over 75% viable compared to the untreated control, respectively. Cells treated with rhamnolipids 222B at 0.01% and 0.009% were ~50% and ~70% viable, respectively. Vero cells treated with rhamnolipids 222A or 222B at or below 0.005% all had over 90% viable. Therefore, rhamnolipids below 0.005% is less toxic to cells during direct contact. To compare rhamnolipids 222B to synthetic surfactant, the toxicity of SDS and Triton X-100 was examined in Hrt18-G cells. As shown in Figure 3B, cells treated with 0.005% of SDS and Triton X-100 had less than 20% viable cells left, which is significantly more toxic than 222B at 0.005%. Therefore, rhamnolipids are less toxic than those synthetic surfactants, such as SDS and Triton X-100.

### 3.3. Comparison of Surfactants against an Enveloped Virus, HSV-1

To prove that viral envelopes are common targets of surfactants, different surfactants were tested against HSV-1 infection in vitro. An ionic surfactant, sodium dodecyl sulfate (SDS), and a non-ionic surfactant, such as Triton X-100, were investigated here. HSV-1 with GFP expression was used to visualize the replication of HSV-1 directly in vitro. Vero cells seeded in 12 well plates were infected with ~1 × 10^5^ PFU/well (i.e., 10^5^ virions per well) in 0.009% of SDS, triton X-100, and 222B, respectively. At 24 hpi, neither HSV-1 CPE nor GFP expression was observed in the presence of SDS or Triton X-100, or 222B (Figure 4). Around 50% and 90% of Vero cells were dead and detached in the presence of SDS, and Triton X-100, respectively. However, no cell death was observed in infected cells in the presence of 222B (Figure 4A, 222B). In line with the observation under the light microscope, no GFP expression was observed in HSV-1 infection in the presence of 222B. GPF expression was only observed in HSV-1only infection without any treatment. This proves SDS and Triton X 100 were too toxic to cells; 222B is much safer biologically than the synthetic surfactants.

### 3.4. Envelope Integrity in the Presence of Surfactants

To visualize the effect of surfactants on viral envelopes directly, HSV-1 virions were treated with different surfactants and examined under the transmission electron microscope (TEM). HSV-1 virions, ~1 × 10^7^ PFU/mL, isolated from both tissue culture supernatant and cell lysates were diluted 1:10 by suspending 0.1 mL of viruses in 0.9 mL of H_2_O containing 0.01% of 222B, SDS, and Triton X-100, respectively, before TEM negative staining. Negative staining is a unique technique of “negative contrast” staining; the contrast is not applied to the object but its environment, using heavy metal salts, such as phosphotungstic acid (PTA). The electron beam can cross biological material easier than the surrounding space. The heavy metal staining forms a dark contrast around viral particles (virions) (Figure 5). The enveloped viruses are those virions with a white ring (white arrow) surrounding the capsid (Figure 5, black arrow). In the untreated viral samples, enveloped virions were visible in each view (Figure 5A, white arrows). However, no enveloped virion was visible in samples treated with 222B (Figure 5B). Likewise, fewer enveloped virions were visible in samples treated with SDS (Figure 5C) or Triton X-100 (Figure 5D). This suggests that virion envelopes had fellen apart in the presence of surfactants. Viral proteins on the envelope are essential for virus entry. If the viruses lose their envelopes, they will not be able to bind to the host receptors, consequently, lose their infectivity.

To further prove that viral envelopes come apart once in contact with rhamnolipids, 100 μL of HSV-1 viruses at ~1 × 10^6^ PFU/mL were diluted in 0.9 mL PBS containing 0.01% of 222B or PBS alone and mixed briefly by vortexing, and immediately inoculated to a 12-well plate seeded with ~1 × 10^5^ Vero cells. Figure 6 shows the cells infected with HSV-1 diluted in PBS with or without 222B. No CPE was observed in cells infected with HSV-1 diluted in PBS containing 0.009% 222B before infection. On the other hand, 80–90 CPE was observed in cells infected with HSV-1 diluted in PBS without 222B, although the culture median contains 0.009% 222B. Similarly, GFP expression was only observed in cells infected with HSV-1 diluted only in PBS and cultured in media containing 0.009% 222B (Figure 6, HSV-1 only). This further demonstrated that rhamnolipids 222B acted directly on the surface of vial envelopes. In addition, it suggests that 0.009% 222B has no effect on HSV-1 replication inside of cells when it is applied to the culture media after HSV-1 infection.

### 3.5. 222B Antiviral Activity against Different Concentrations of HSV-1

To determine the potency of 222B against enveloped viruses, different concentrations of HSV-1 were treated with different concentrations of 222B before inoculation. The viruses were diluted in different concentrations of 222B first, then titrated immediately by standard plaque assay. Therefore, HSV-1 viruses were only exposed to 222B treatment during the dilution and inoculation stages. The plaque assay was performed in 12-well plates, where only 0.1 mL diluted viruses were inoculated to each well. Following 1 h absorption at 37 °C, 2 mL of overlay media was added to each well. As shown in Figure 7, no viral plaque was observed in wells infected with ~2 × 10^6^ PFU HSV-1 diluted in 0.009% 222B. Similarly, 222B at 0.0045% reduced the input virus titer from ~2 × 10^6^ PFU to ~2 × 10^2^ PFU, ~2 × 10^5^ PFU to 10–20 PFU, ~2 × 10^4^ PFU to a few PFU, respectively. 222 B at 0.001% reduced the input virus down to about 10-fold. However, 222B at a concentration below 0.001% had little or no antiviral activity. These results demonstrate that 222B at 0.0045% and 0.009% can inactivate ~1 × 10^4^ and ~1 × 10^6^ PFU of enveloped viruses, respectively.

### 3.6. Contact Inhibition against HSV-1 on 222B Coated Plastic Surface

If a biologically safe antiviral product remains active on a coated surface, we could use it to kill the enveloped viruses and prevent the spread of respiratory viruses, such as SARS-CoV2. To this end, 222B anti-HSV-1 activity on the plastic surface was investigated here. The surface of a 48-well plate, ~1 cm^2^ per well, was coated with 20 μL of 222B at 0.005% and air-dried before testing. Viral droplets in 10 μL containing ~2 × 10^4^ or ~2 × 10^3^ PFU of HSV-1 was directly applied to the 222B coated surface. The contact between the coated surface and viruses was tested for 1 or 5 min and then stopped by adding 1 mL of PBS elution buffer. The post-contact viruses were eluted at room temperature for 1 h on a plate rocker. If the rhamnolipids can inactivate envelopes on the coated surface, then no infectious viruses or reduced infectious viruses will be eluted. Figure 8 shows the number of HSV-1 PFU detected after direct contact. When ~2 × 10^4^ PFU HSV-1 in 10 μL was applied to the 222B coated plastic surface, around 200 PFU was detected following 1-min contact, and about 10 PFU was detected following 5-min contact (Figure 8A). Likewise, when 2 × 10^3^ PFU was applied to the 222B coated plastic surface, about 30 PFU on average were detected in the elution media following 1-min contact; no PFU was detected following 5-min contact (Figure 8B). On the other hand, when ~2 × 10^4^ or ~2 × 10^3^ PFU HSV-1 in 10 μL was applied to the PBS coated surface, similar numbers of PFU were recovered (Figure 8, PBS). The reduction at 5 min contact (222B-5 min) is statistically significant compared to untreated control (*p* < 0.0006 for 2 × 10^4^ PFU HSV-1 per droplet, and *p* < 0.0002 for 2 × 10^3^ PFU per droplet). This suggests that rhamnolipids on the plastic surface can kill enveloped viruses.

### 3.7. Contact Inhibition against HSV-1 on 222B Coated Fabric Surface within 3–5 min

To determine whether 222B on the coated surgical mask can inactivate enveloped viruses, about 1-cm^2^ mask fabrics were coated with 50 μL of 0.005% 222B and air-dried before testing. The viral droplets in 50 μL were applied directly to the 222B coated mask fabric for 3–5 min. The interaction between viruses and 222B on the fabric surface was stopped by adding 1 mL PBS elution buffer (which will dilute 222B to a concentration with little or no antiviral activity). The post-contact viruses were eluted in 1 mL PBS at room temperature for 1 h on a plate rocker. When 1.45 × 10^4^ PFU per droplet was applied to the 222B coated mask fabric, less than 10 PFU on average were detected in the elution media (Figure 9A). When 1.45 × 10^3^ PFU per droplet was applied to the 222B mask fabric, no viral plaque was detected in the elution media (Figure 9B). In contrast, when 1.45 × 10^4^ PFU per droplet was applied to the 222B coated fabric after application of 1 mL elution media without direct contact, about 2.9 × 10^3^ PFU was detected in the elution media (Figure 9A, 222B-NC). Likewise, when 1.45 × 10^3^ PFU per droplet was applied to the coated fabric without direct contact, about 2.2 × 10^2^ PFU was detected in the elution media (Figure 9B, 222B-NC). The reduction from direct contact (222B) is statistically significant compared to non-direct contact (222B-NC) (*p* < 0.0001 1.45 × 10^4^ PFU HSV-1 per droplet, and *p* < 0.0001 for 2.9 × 10^3^ PFU per droplet). When input viruses at 1.45 × 10^4^ PFU and 1.45 × 10^3^ PFU per droplet were applied to the PBS coated fabrics, about 2.2 × 10^3^ PFU/mL and 2.2 × 10^2^ PFU/mL were detected in the elution media, respectively (Figure 9). These results demonstrated that 0.005% of 222B coated on the mask could inactivate about ~1 × 10^3^ PFU enveloped viruses per cm^2^.

## 4. Discussion

Rhamnolipids are biosurfactants and have many advantages over synthetic surfactants, such as low toxicity and high biodegradability. In this study, 222B made of predominant monorhamnolipids were found to have better anti-HSV-1 activity than that of predominant dirhamnolipid, 222A (Figure 1). At the same concentration, such as 0.009%, 222B killed more HSV-1 viruses than 222A did (Figure 1, HSV-1+222B). This difference between 222A and 222B could be that 222A has an extra rhamnose sugar group (rha), which may interfere with the interactions between the lipid tail with the phospholipid bilayers of envelopes. Therefore, the monorhamnolipids are better anti-enveloped viral agents.

Like synthetic surfactants, rhamnolipids are toxic to cells when their concentration is at or above 0.05% (Figure 3). However, comparing to SDS (anionic surfactant) and Triton X-100 (a non-ionic surfactant), rhamnolipids are less toxic at lower concentrations, such as 0.009% and 0.005%, which are still potent against enveloped viruses. Interestingly, 222A is slightly less toxic to Vero cells than 222B when the concentration is below 0.005% (Figure 3A). This may come from the difference in their structures, where 222A is composed of 95% of dirhamnolipids, while 222B is composed of 95% of monorhamnolipids. In line with this observation, a similar product 223A, containing 90% of dirhamnolipids, is also less toxic than 223B consist of 90% of monorhamnolipids.

The primary action of rhamnolipids against viruses was found to be specific to the envelopes. In the presence of rhamnolipids, such as 222B, viral envelopes became unstable and fell apart from the capsids. As shown in Figure 5, HSV-1 virion treated with 222B had no envelopes left around the capsids. Also, dense spots on the naked capsids were observed in virions treated with 222B (Figure 5B). This suggests that heavy metal staining had gone inside of the capsids. This could mean that the viral capsids were also compromised in the presence of 222B, which suggests that rhamonlipids may have antiviral activities to the non-enveloped viruses too. Although non-enveloped virions were also visible in the untreated sample (Figure 5A), virions, used here, were isolated from tissue culture supernatant and cell lysates, where there are usually many incomplete virions.

The action of rhamnolipids against the envelopes was found to be pretty fast and instantaneous. The time that it takes to dilute the virus samples is between 5–10 min. The diluted viruses were then inoculated to wells seeded with cells immediately. We found that viruses at about ~1 × 10^6^ PFU diluted in 0.009% 222B lost all their infectivity; produced no CPE in the infected cells (Figure 6, HSV-1+222B). However, when the viruses (also ~1 × 10^6^ PFU) were inoculated first without diluting in 222B, and the infection was cultured in media containing 0.009% 222B, HSV-1 replication was not affected (Figure 6, HSV-1 only). These results suggest that 222B damaged the viral envelops during the dilution phase when they directly contacted each other, and rhamnolipids could destabilize the viral envelope within 5–10 min and destroy their infectivity. Once the viruses enter the cell, rhamnolipids have no effect on virus replication inside of the infected cells.

In this study, rhamnolipids’ antiviral activity was investigated mostly in concentrations that are less toxic to the cell, such as 0.009% and 0.005%. Although 0.009% is relatively low, 222B at this concentration can kill ~1 × 10^7^ PFU/mL of HSV-1 or BCoV. Also at 0.0045%, 222B can kill ~2 × 10^5^ PFU/mL (Figure 7). Therefore, the rhamnolipid anti-enveloped virus activity is potent and dose-dependent; the higher the concentration is, the higher the killing activity is. Not only 222B can inactivate enveloped virus in the liquid phase, but it also can kill viruses on the coated surface. As results shown in Figure 8 and Figure 9, 222B on coated plastic and fabric surface could kill HSV-1 applied to the surface directly. 005% of 222B on the mask and plastic surface is capable of killing at least ~1 × 10^3^ PFU HSV-1/cm^2^ within 5 min (Figure 8), 0.009% of 222B on the coated mask and plastic surface can kill almost ~1 × 10^5^ PFU HSV-1/cm^2^. Therefore, the killing capacity on the coated surface is also dose-dependent, where the higher the concentration is, the higher the killing activity is.

Respiratory viruses are transmitted mostly via droplets or aerosols, or body secretions, such as tears, nasal or oral discharges. The number of viruses shed in the nasal or oral secretion can range from 10^4^ to 10^7^ genome copies/mL (or 10^2^ to 10^5^ genome copies/10 uL) depending on the severity of the infection and the types of viruses [33,34]. For example, Middle East respiratory syndrome coronavirus (MERS-CoV) is a highly pathogenic coronavirus, can cause 34% mortality in infected humans [35]. Almost 5 × 10^6^ genome copies/mL can be shed in sputum from patients with severe MERS [36]. Severe acute respiratory syndrome coronavirus 2 (SARS-CoV-2) is the virus responsible for the 2019 Coronavirus disease (COVID-19) pandemic. It was estimated that 10^4^–10^7^ genome copies/mL could be shed in nasal and pharyngeal specimens from COVID-19 patients [34]. Around 10^4^–10^5^ genome copies/mL SARS-CoV2 could be shed in sputum from severely infected COVID-19 patients [37,38]. In the case of Influenza A virus infection, 10^4^–10^7^ virions/mL could be shed from patients with uncomplicated acute influenza A [33]. If these respiratory viruses came out in a 10 μL droplet size, there would be 10^2^–10^5^ virions per droplet from the infected patient. If the infected person wore rhamnolipid coated masks, almost all those viruses would be killed inside the mask coated with 0.005% 222B. Our contact study was performed on only ~1 cm^2^ surface. A regular mask will have a large surface, which could have more kill capacity than we observed within a 1 cm^2^ surface. A mask coated with either 0.005% or 0.009% 222B would eliminate most of the viruses shed from an acutely infected person, which could limit the emission of viruses from the infected person and stopping the spread of diseases. Rhamnolipids are very stable on the coated surface. 222B coated on plastic remained active for as long as six months.

We have lived with the COVID-19 pandemic for almost a year, where mask-wearing, social distancing, and business shutdown have not entirely stopped the spread of the disease. The key to preventing transmission is eliminating the emission of SARS-CoV2 from an infected person and stopping the virus from propagating in the human population. The best way to stop emissions is to prevent an infected person from shedding infectious viruses to the environment. If we could have all the COVID-19 patients wearing 222B coated masks, 222B could kill most of the viruses shed in nasal or oral secretions and prevent the spread of this disease. During quarantine, if we would all use 222B coated masks, it could break the transmission chain and end the infection within short time, which will prevent the business and school from shutting down we experienced in 2020. The widespread use of 222B coated masks could end the COVID-19 pandemic much sooner if we implement masks that can kill viruses. Therefore, there is an urgent need to apply rhamnolipids or similar products to masks or make 222B coated masks available to the public to stop the pandemic of COVID-19.

## 5. Conclusions

In summary, this study demonstrated that rhamnolipids have antiviral activity against enveloped viruses. Many respiratory viruses are enveloped viruses, such as SARS-CoV2, influenza viruses. Rhamnolipids can kill enveloped viruses in solution (liquid phase) and on the coated surface. Direct contact with rhamnolipids can destroy the envelopes within five minutes. Rhamnolipids, 222B at 0.005%–0.009% can kill ~1 × 10^4^–1 × 10^6^ PFU/mL of enveloped viruses in solution, ~1 × 10^3^–10^5^ PFU/cm^2^ enveloped viruses on ~1 cm^2^ surface, respectively. The application of rhamnolipids 222B on masks could help us stop the emission of viruses from the infected person, break the transmission chain, and end the pandemic of COVID-19 in weeks.

## Figures and Tables

**Figure 1 viruses-13-00322-f001:**
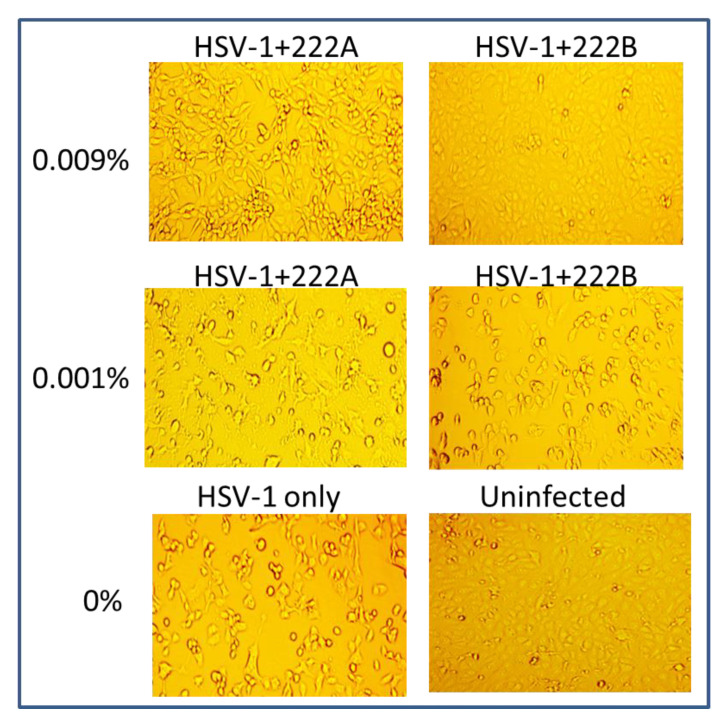
HSV-1 infected Vero cells in the presence or absence of rhamnolipids. The concentration of rhamnolipids is indicated on the left of the images. HSV-1 infection with 222A or 222B is labeled above the image as HSV-1+222A or HSV-1+222B. Vero cells were infected with ~2 × 10^6^ FPU/well, and the images were taken at 24 hpi at 10× magnification.

**Figure 2 viruses-13-00322-f002:**
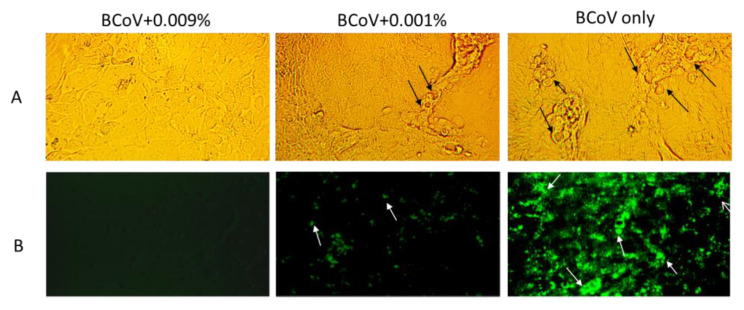
Bright light images and direct immunofluorescence staining of BCoV infected cells in the presence and absence of 222B. **A**: Hrt18G cells infected with ~1 × 10^6^ TCID_50_ of BCoV per well in a 12-well plate. The concentration of 222B is labeled above the images in the BCoV infection. The bright light images were captured at 72hpi at 20× magnification. Black arrows indicate vesicles within the virus-infected cells. **B**: Hrt18G cells at 72hpi were stained with 1:40 diluted BCoV antibody conjugated with FITC and viewed under the fluorescence microscope. Fluorescent images were captured at 10× magnification. White arrows indicate FITC stained BCoV antigens within the infected cells.

**Figure 3 viruses-13-00322-f003:**
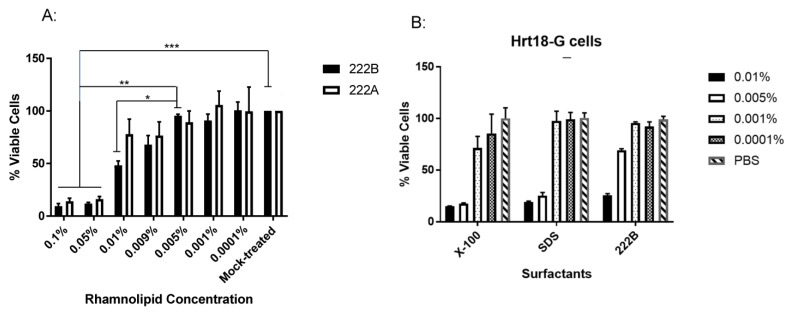
The cytotoxicity of Rhamnolipids in vitro. Vero cells (**A**) or Hrt-18 G cells (**B**) were incubated with the indicated concentration of rhamnolipids or surfactants in PBS for 1 h. The treatment was then removed, and the cells were washed twice with PBS and further incubated for 24 h in EMEM with 5% serum and antibiotics as described in materials and methods. Cell viability was evaluated with XTT cell viability kit III (PromoKine), which measures metabolic activity based on the extracellular reduction of XTT by NADH produced in the mitochondria, and expressed as a percentage of the Mock-treated control (*n* = 3). A significant statistical difference from the PBS-treated control (Mock-treated) or 0.005% 222B (A) is marked with * *p* < 0.05, ** *p* < 0.001, *** *p* < 0.0001 (two-way ANOVA with Bonferroni post-test).

**Figure 4 viruses-13-00322-f004:**
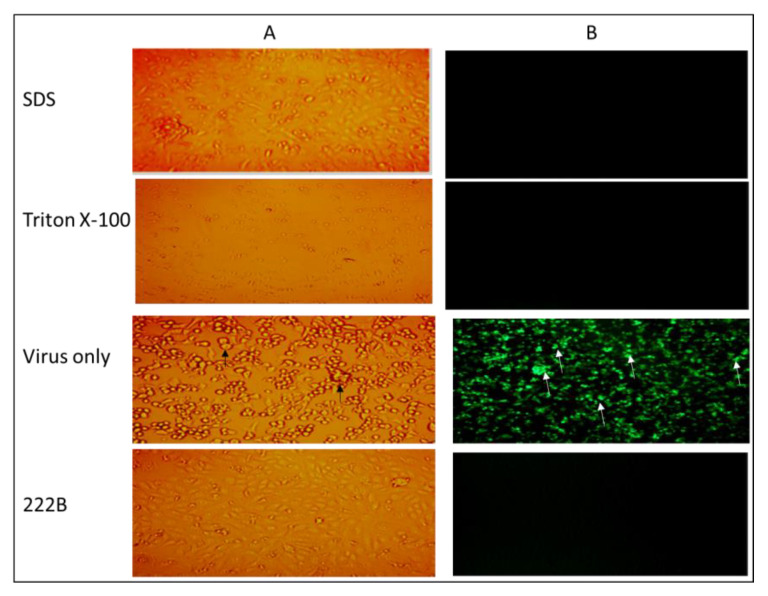
Bright light (**A**) and fluorescence (**B**) images of Vero cells infected with HSV-1-GPF. HSV-1-GPF at ~1 × 10^5^ PFU/well was inoculated to Vero cells in the presence of SDS, Triton X-100, 222B, and HSV-1 only, respectively. All surfactants were made at 0.009%% in the inoculants. A: The bright light image was captured at 10× magnification. B: Fluorescent images were captured at 20× magnification. Black arrows indicate CPE under the light microscope, such as cell rounding and aggregation in HSV-1-infected cells. White arrows indicate GFP expression from GFP-HSV-1 replication under the fluorescent microscope.

**Figure 5 viruses-13-00322-f005:**
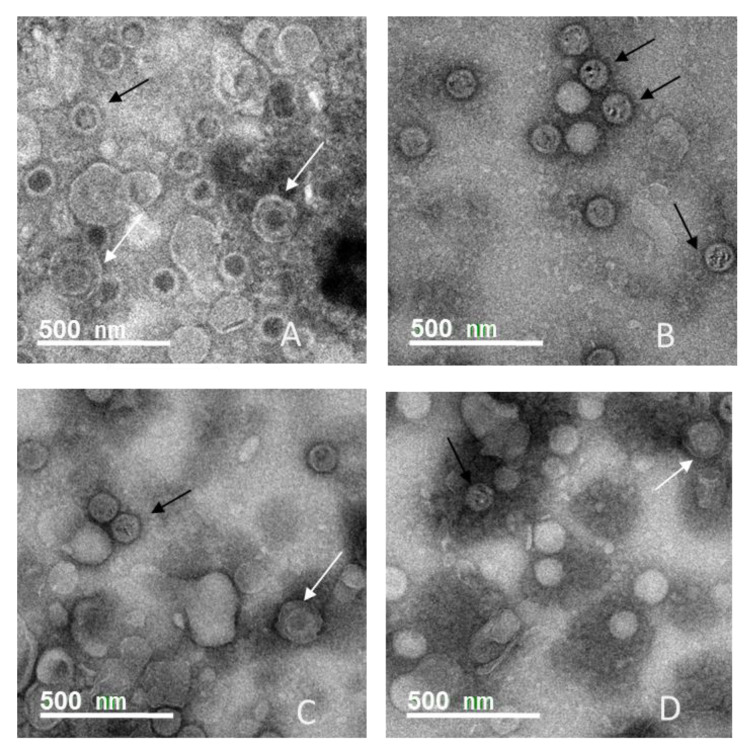
Electron micrograph images of HSV-1 virions treated with different surfactants. The images were HSV-1 virions without treatment (**A**), virions treated with 0.009% of 222B (**B**), 0.009% of SDS (**C**), and 0.009% of Triton X-100 (**D**), respectively. The black arrow indicates the capsids of virions without envelopes. The white arrow indicates the envelope of the virion. Scale bar = 0.5 μm.

**Figure 6 viruses-13-00322-f006:**
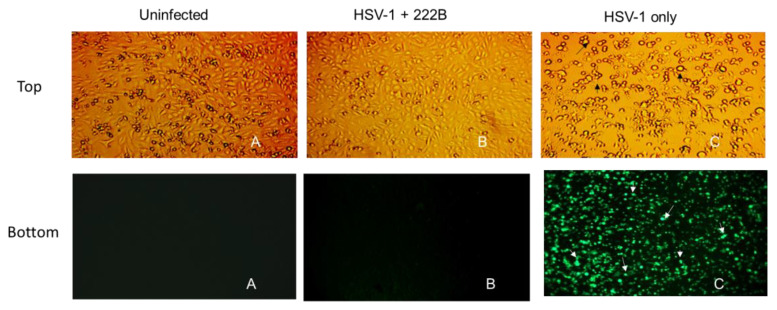
Bright light (Top) and fluorescence (bottom) images (10× magnification) of HSV-1-GFP infection. “HSV-1+222B” represents cells infected with HSV-1 that were diluted with PBS containing 0.009% 222B and cultured in a media without 222B. “HSV-1 only” represents cells infected with HSV-1 that were diluted with PBS only and cultured in a media containing 0.009% 222B. Both bright light and fluorescent images were captured at 10× magnification. Black arrows indicate CPE, such as cell rounding and aggregation in HSV-1-infected cells. White arrows indicate GFP expression from GFP-HSV-1 replication.

**Figure 7 viruses-13-00322-f007:**
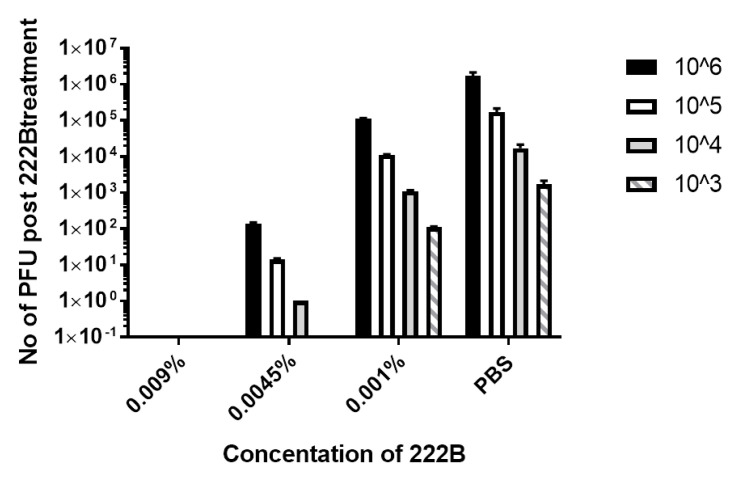
Plaque reduction of 222B against HSV-1. HSV-1 stock at ~2 × 10^7^ PFU/mL was diluted to ~2 × 10^6^ PFU/mL (solid bar), ~2 × 10^5^ PFU/mL (open bar), ~2 × 10^4^ PFU/mL (grey bar), ~2 × 10^3^ PFU/mL (stripped bar) in PBS containing 222B at 0.009%, or 0.005%, or 0.001% in the final concentration, respectively. The diluted viruses were then titrated in a 12-well plate seeded with Vero cells. Plaque formation was quantified at 4–5 dpi and expressed as average PFU post 222B treatment (*n* = 3).

**Figure 8 viruses-13-00322-f008:**
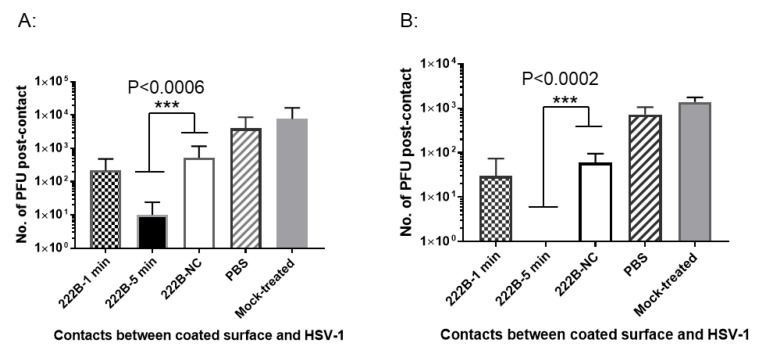
Contact inhibition against HSV-1 on the 222B coated plastic surface. The plastic surface was exposed to input viruses at ~2 × 10^4^ PFU (**A**) or ~2 × 10^3^ PFU (**B**) per well. The X-axis stands for contacts between coated surface and HSV-1; the Y-axis represents the number of PFU post-contact. The checked bar stands for direct contact between 222B and HSV-1 for 1 min (222B-1 min); the solid bar stands for direct contact between 222B and HSV-1 for 5 min (222B-5 min); the open bar stands for the no direct contact between HSV-1 and 222B (222B-NC), the striped bar stands for direct contact between PBS coated surface and HSV-1 for 5 min (PBS); and the gray bar stands for HSV-1 diluted in PBS directly (Untreated). The viruses eluted after direct contact were quantified by plaque assay as average PFU post 222B treatment (*n* = 3). A significant statistical difference from the 222B-NC is marked with *** *p* < 0.0002 and *p* < 0.0006 (two-way ANOVA with Bonferroni post-test).

**Figure 9 viruses-13-00322-f009:**
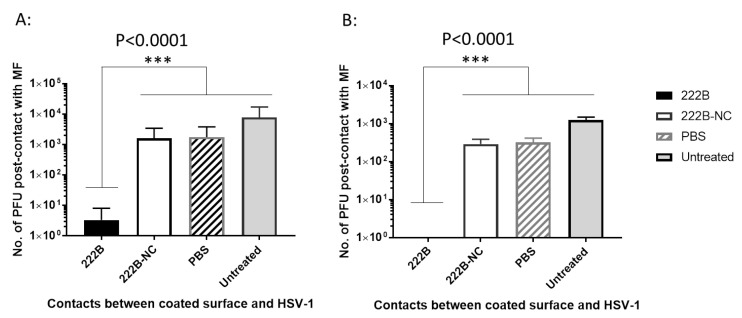
Contact inhibition against HSV-1 on the 222B coated fabric surface within 3–5 min. The mask fabric surface was exposed to input viruses at ~1.45 × 10^4^ PFU (**A**) or ~1.45 × 10^3^ PFU (**B**) per cm^2^. The X-axis stands for contacts between coated surfaces and HSV-1; the Y-axis represents the number of PFU post-contact with the coated fabrics. The solid bar stands for direct contact between 222B and HSV-1 (222B); the open bar stands for the no direct contact between HSV-1 and 222B (222B-NC); striped bar stands for direct contact between PBS coated surface and HSV-1 (PBS), and the gray bar stands for HSV-1 diluted in PBS directly (Untreated). The viruses eluted after direct contact were quantified by standard plaque assay as average PFU post 222B contact (*n* = 3). A significant statistical difference from the 222B-NC and PBS control is marked with *** *p* < 0.0001 (two-way ANOVA with Bonferroni post-test).

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
