# Peer review of "Application of Environment-Friendly Rhamnolipids against Transmission of Enveloped Viruses Like SARS-CoV2"

_viruses, 2021, doi:10.3390/v13020322_

Round 1

Reviewer 1 Report

The Authors carried out several assays to evaluate the antiviral ability of two rhamnolipids against eneveloped viruses such as HSV-1 and BCoV in the perspective of using these compounds as antiviral compounds on protective masks for another envelopeded viruses such as Sars-CoV2.

The results are very interesting being the two rhamnolipids tested (222A and 222B) able to kill both the viruses in 5 minutes and not cytoxic at the concentration of 0,0045 and 0,009%. The activity of the compounds results to be expressed on virus’s envelope.

The manuscript is well written and very interesting in the view of Sars-CoV2 stopping transmission Nevertheless, few minor point should be addressed prior to publication:

  • Page 3, line 142: “…with HSV-1 at each concentration…”. Which concentration?. Please explain
  • Page 4 lines 156-157: “… without contacting the coated surface…” Please explain or rephrase. The sentence could not be clear for the readers
  • Page 4 lines 169-170: “…without contacting the fabrics…” Please explain or rephrase. The sentence could not be clear for the readers
  • there are several mispelling/mistyping throughout the manuscript (page 9, line 313: “3.5.222. B activity….”; elusion instead of elution on page 10, somewhere in lines 373-385; i.e COVD instead of COVID on page 12 lines 459, 474;)
  • some characters should be written in subscript (i.e TCID50 instead of TCID50 and CO2 instead of CO2, somewhere in methods section)

Author Response

Reviewer 1

The Authors carried out several assays to evaluate the antiviral ability of two rhamnolipids against eneveloped viruses such as HSV-1 and BCoV in the perspective of using these compounds as antiviral compounds on protective masks for another envelopeded viruses such as Sars-CoV2.

The results are very interesting being the two rhamnolipids tested (222A and 222B) able to kill both the viruses in 5 minutes and not cytoxic at the concentration of 0,0045 and 0,009%. The activity of the compounds results to be expressed on virus’s envelope.

The manuscript is well written and very interesting in the view of Sars-CoV2 stopping transmission Nevertheless, few minor point should be addressed prior to publication:

    Page 3, line 142: “…with HSV-1 at each concentration…”. Which concentration?. Please explain

RESPONSE: There were several concentrations of HSV-1 tested in HSV-1 plaque reduction assay, which ranged from 1X103 t o1X106.  To make it clear, now we have revised this sentence to “ HSV-1 at different concentrations, ranging from 1X103 to 1X107 PFU/ml”.

    Page 4 lines 156-157: “… without contacting the coated surface…” Please explain or rephrase. The sentence could not be clear for the readers

RESPONSE: This specific sentence has now been revised. The non-surface contact controls were 222B coated wells, the viruses in 10 µl were added to the coated wells after application of 1 ml of PBS elution buffer; therefore, they had no direct contacts with the coated surface.

    Page 4 lines 169-170: “…without contacting the fabrics…” Please explain or rephrase. The sentence could not be clear for the readers

RESPONSE: The related sentence is now revised as the following: The non-surface contact controls were 222B coated fabrics, the viruses in 10 µl were added onto the coated fabrics after 1 ml of PBS elution buffer, therefore, had no direct contact with the coated fabrics.

    there are several mispelling/mistyping throughout the manuscript (page 9, line 313: “3.5.222. B activity….”; elusion instead of elution on page 10, somewhere in lines 373-385; i.e COVD instead of COVID on page 12 lines 459, 474;)

RESPONSE: Agree. Corrections have been made in multiple places.

    some characters should be written in subscript (i.e TCID50 instead of TCID50 and CO2 instead of CO2, somewhere in methods section)

RESPONSE: Agree. Correction has been made.

Reviewer 2 Report

The manuscript by Jin, Black, and Sawyer was a relatively comprehensive research article on the potential treatment effects of viruses. The authors focused on the viral envelopes and tested the rhamnolipids effects using cell lines and pseudotyped viruses. Experiments were well performed and data was well collected. There were some significant concerns:

  • Inconsistent design significantly reduced the quality of research. Specifically, for example for incubation time, the cell viability assay in section Rhamnolipid cytotoxicity assay in vitro, Page 3, cells were treated with 222A or 222B “for 1 hr” (Line 95) and “further incubated for 24 h” (Line 97); following 1-hr absorption for only 222B, “At 24 or 72 h post-infection” (Lines 112-113); infected flacks were harvested “at 3-4 days post-infection” (Line 126); “plaques were counted at 4-5 dpi” (Line 146); virus droplets “were added to the bottom of the well for 1 min or 5 min” (Line 152); virus droplets “were added to the coated fabrics for 3-5 min” (Line 165). Please justify the differences. Please provide additional data for comparison if needed.
  • Time course study is needed. Data provided at least for viruses with GFP for tracking experiments.
  • The Fabric coating (Methods in Page 4) and anti-viral effects should be repeated with cells.
  • Section 2. Cytotoxicity of rhamnolipids in Vitro and related experiments, the rationale of the cell viability or “cytotoxicity” test was unclear.
  • Line 287, “will not be able to bind to the host receptors” were hypothesized by the authors; however, there was no matched experiment or data. It was understood that electromicroscopy was utilized to confirm the binding. But no quantified data presented.

Author Response

The manuscript by Jin, Black, and Sawyer was a relatively comprehensive research article on the potential treatment effects of viruses. The authors focused on the viral envelopes and tested the rhamnolipids effects using cell lines and pseudotyped viruses. Experiments were well performed and data was well collected. There were some significant concerns:

  • Inconsistent design significantly reduced the quality of research. Specifically, for example for incubation time, the cell viability assay in section Rhamnolipid cytotoxicity assay in vitro, Page 3, cells were treated with 222A or 222B “for 1 hr” (Line 95) and “further incubated for 24 h” (Line 97); following 1-hr absorption for only 222B, “At 24 or 72 h post-infection” (Lines 112-113); infected flacks were harvested “at 3-4 days post-infection” (Line 126); “plaques were counted at 4-5 dpi” (Line 146); virus droplets “were added to the bottom of the well for 1 min or 5 min” (Line 152); virus droplets “were added to the coated fabrics for 3-5 min” (Line 165). Please justify the differences. Please provide additional data for comparison if needed.

RESPONSE:

“the cell viability assay in section Rhamnolipid cytotoxicity assay in vitro, Page 3, cells were treated with 222A or 222B “for 1 hr” (Line 95) and “further incubated for 24 h” (Line 97); following 1-hr absorption for only 222B”,--

These are experiments to assess the rhamnolipid's toxicity directly in tissue cultures, ie. Vero cells and Hrt18-G cells. Viruses can only grow in viable cells, if the rhamnolipids are toxic to cells, they will kill cells, and no viable cells will be left for viruses to grow. Therefore, the cytotoxicity of rhamnolipids needed to be tested before we can evaluate their antiviral activity. We can only measure rhamnolipids' antiviral effect in conditions where they do not interfere with virus infection in vitro. In this experiment, the rhamnolipids are added directly to Vero cells or Hrt18-G cells and incubated with these cells for 1 hr. The one hour exposure time is commonly used in viability assay to measure the tested compounds' toxicity. Following 1 hour incubation, the cells usually are cultured for 24h before the XXT kit can measure viability.

“ At 24 or 72 h post-infection” (Lines 112-113)”, --

The incubation time used here is the minimum time needed to see the cytopathic effect (CPE) of HSV-1 infection or BCoV infection under the light microscope. CPE from HSV-1 infection in Vero cells is visible at 24h post-infection, while BCoV CPE is visible at 72h post-infection. That is why two different incubation time was used to exam the two different enveloped viruses.

“infected flacks were harvested “at 3-4 days post-infection” (Line 126);”

These are the production of HSV-1 virions to be used for TEM examination in Fig. 5. HSV-1 infection was given the maximum time to grow in the infected cell to ensure enough virions are produced.

“plaques were counted at 4-5 dpi” (Line 146);”

These are standard plaque assays to measure the HSV-1 titer. These assays will give the viruses the maximum time to grow in vitro. It normally takes 4-5 days to allow individual virus spreading to adjacent cells and forms visible lesions. The number of days used here does not change the outcome of the result, but it does make it easier to count.

 “virus droplets “were added to the bottom of the well for 1 min or 5 min” (Line 152)”.

These experiments are to measure how fast the rhamnolipids can kill the enveloped viruses upon contact. In this experiment, viruses were added to a plastic surface coated with 222B. To measure the time needed to destroy the enveloped viruses, the virus droplet was allowed to sit on the 222B coated surface for 1 min or 5 min. The contact was stopped by adding 1 ml PBS to elute the virus from the surface.

 “virus droplets “were added to the coated fabrics for 3-5 min” (Line 165). ”

These experiments tested whether the rhamnolipid on coated fabric surface can kill enveloped viruses within 3-5 min. In this experiment, viruses were added to mask fabric coated with 222B, and virus droplet was allowed to sit on the 222B coated mask surface for 3-5 min. The contact between the virus and 222B was stopped by adding 1 ml PBS to elute the virus from the surface.

  • Time course study is needed. Data provided at least for viruses with GFP for tracking experiments.

RESPONSE: GFP-HSV-1 expression can only be seen at least 12h post-infection. The time point used here is in line with the HSV-1 replication cycle at 24h post-infection when CPE is visible under the light microscope. These are qualitative assay where you either see the effect (CPE) or do not see the effect (CPE) from virus infection.

The Fabric coating (Methods in Page 4) and anti-viral effects should be repeated with cells.

RESPONSE: Rhamnolipids can damage the viral envelops of HSV-1 directly in 5-10 min, shown in Fig. 5. This study also demonstrated that viruses were inactivated in a dilution medium containing rhamnolipids, which is shown in Fig 6B. All the results demonstrated that the rhamnolipids anti-viral effect is via direct contacts.

Section 2. Cytotoxicity of rhamnolipids in Vitro and related experiments, the rationale of the cell viability or “cytotoxicity” test was unclear.

RESPONSE: See above. Viruses can only grow in viable cells. If the rhamnolipid is toxic to cells, it will interfere with virus infection in tissue culture. Therefore, cytotoxicity of rhamnolipids needs to be examined before its antiviral effect can be measured in tissue culture.

Line 287, “will not be able to bind to the host receptors” were hypothesized by the authors; however, there was no matched experiment or data. It was understood that electromicroscopy was utilized to confirm the binding. But no quantified data presented.

RESPONSE: Enveloped viruses have glycoproteins on their surface of envelopes. These glycoproteins are host receptor binding proteins. If enveloped viruses lose their envelopes, they will lose their glycoproteins. Therefore, viruses without the envelopes can not bind to the host receptors and infect their target cells. This is demonstrated in Figure 6, where viruses diluted in rhamnolipids fail to infect the Vero cells (Fig. 6B). In contrast, viruses diluted in the absence of rhamnolipids can infect the cell and produce CPE and GFP despite the tissue culture media containing the same concentration of rhamnolipids.